# Reliability and Validity Examination of a New Gait Motion Analysis System

**DOI:** 10.3390/s25041076

**Published:** 2025-02-11

**Authors:** Tadamitsu Matsuda, Yuji Fujino, Tomoyuki Morisawa, Tetsuya Takahashi, Kei Kakegawa, Takanari Matsumoto, Takehiko Kiyohara, Hiroshi Fukushima, Makoto Higuchi, Yasuo Torimoto, Masaki Miwa, Toshiyuki Fujiwara, Hiroyuki Daida

**Affiliations:** 1Department of Physical Therapy, Faculty of Health Science, Juntendo University, Tokyo 113-8421, Japan; y.fujino.pb@juntendo.ac.jp (Y.F.); t.morisawa.ul@juntendo.ac.jp (T.M.); te-takahashi@juntendo.ac.jp (T.T.); fhs8323006@juntendo.ac.jp (K.K.);; 2Global Development Center, Development Department, Development Section, IMASEN Electric Industrial Co., Ltd., Inuyama 484-0083, Japantakehiko_kiyohara@imasen.co.jp (T.K.); hiroshi_fukushima@imasen.co.jp (H.F.); makoto_higuchi@imasen.co.jp (M.H.); yasuo_torimoto@imasen.co.jp (Y.T.); masaki_miwa@imasen.co.jp (M.M.); 3Department of Rehabilitation Medicine, Graduate School of Medicine, Juntendo University, Tokyo 113-8421, Japan; t-fujiwara@juntendo.ac.jp

**Keywords:** motion analysis, gait, validation, human pose estimation

## Abstract

Recent advancements have made two-dimensional (2D) clinical gait analysis systems more accessible and portable than traditional three-dimensional (3D) clinical systems. This study evaluates the reliability and validity of gait measurements using monocular and composite camera setups with VisionPose, comparing them to the Vicon 3D motion capture system as a reference. Key gait parameters—including hip and knee joint angles, and time and distance factors—were assessed under normal, maximum speed, and tandem gait conditions during level walking. The results show that the intraclass correlation coefficient (ICC(1,k)) for the 2D model exceeded 0.969 for the monocular camera and 0.963 for the composite camera for gait parameters. Time–distance gait parameters demonstrated excellent relative agreement across walking styles, while joint range of motion showed overall strong agreement. However, accuracy was lower for measurements during tandem walking. The Cronbach’s alpha coefficient for time–distance parameters ranged from 0.932 to 0.999 (monocular) and from 0.823 to 0.998 (composite). In contrast, for joint range of motion, the coefficient varied more widely, ranging from 0.826 to 0.985 (monocular) and from 0.314 to 0.974 (composite). The correlation coefficients for spatiotemporal gait parameters were greater than 0.933 (monocular) and 0.837 (composite). However, for joint angle parameters, the coefficients were lower during tandem walking. This study underscores the potential of 2D models in clinical applications and highlights areas for improvement to enhance their reliability and application scope.

## 1. Introduction

Aging often brings changes in joint range of motion (ROM) that can lead to noticeable gait irregularities, especially in individuals with conditions such as osteoarthritis, stroke, or Parkinson’s disease [1]. Consequently, clinical gait analysis has become essential for effective rehabilitation, offering valuable insights into patient mobility and overall health. Standard clinical assessments [2] commonly involve timed [3] and shuttle [4] walking tests, during which a patient walks a distance of 10 m or less while parameters such as step length, step width, gait velocity, step count, cadence, gait cycle time, step duration, and kinematic joint angles are measured. To ensure accuracy, clinicians often rely on high-cost motion capture systems that apply a combination of machine learning and manual annotations. However, these systems incur high costs, are complex, and require a level of operating expertise, limiting accessibility for many patients. Those who do participate must endure repetitive visits to specialized laboratories, the placement of anatomical markers, and, in some cases, complex and repetitive imaging processes, which can alter gait mechanics leading to inaccurate measurements and increased patient burden [5,6].

Current state-of-the-art clinical systems include Vicon (Vicon Nexus2; Vicon Motion Systems Ltd., Oxford, UK) and Optitrak [7,8]. Vicon, the current gold standard for motion capture, is widely regarded for its accuracy, producing a high level of reliability and internal statistical validity. “Internal validity” is essentially a guarantee that the processes used to make quantitative assessments are sound, repeatable, and confident at a very high level (e.g., 95–98%, depending on the measure). For Vicon, this guarantee refers to its function of making predictions based on three-dimensional (3D) video input. Therefore, it not only handles stereoscopic image data well, but also keeps track of features across the sequential frames of any given video. However, Vicon requires the placement of anatomical markers on the patient.

Recently, markerless systems using accelerometers or wearable data collection units have emerged as alternatives to traditional marker-based systems, and discussions have been held regarding their reliability and validity [9,10]. However, these systems are generally less reliable when capturing detailed kinematic data and present significant operational challenges. Despite these issues, markerless systems offer unique advantages. For example, they have the potential to reduce the time and costs associated with marker-based approaches [11]. Additionally, these systems capture motion directly from the subject’s body surface, reducing errors caused by skin movement artifacts and are less susceptible to inaccuracies resulting from marker misalignment or movement, which are common drawbacks in marker-based methods [12,13,14,15]. A significant breakthrough in markerless technology came with the release of Microsoft’s Kinect motion-sensing device in 2010. The Kinect is known for its ease of use, portability, and affordability. Various studies investigating gait analysis with Kinect-based systems have, however, shown significant differences in accuracy compared to Vicon. To address these discrepancies, researchers analyzed the Kinect’s internal validity [12] by comparing its measurements of an 8-m walking test to those obtained from Vicon. The findings revealed that the Kinect’s intraclass correlation coefficients (ICCs) were reliable only at very low data acquisition rates, limiting its clinical applicability. Subsequent studies with the Kinect V2, which uses an automated body tracking algorithm, have shown promise in assessing temporal gait parameters. However, while the Kinect V2 can track basic lower body movements, its reliability and validity for detailed kinematic analysis remain low [16,17,18]. Therefore, although Kinect may struggle with high-accuracy kinematic data, research suggests it could still be a valuable tool for capturing broad spatial and temporal parameters of gait [19].

VisionPose is a markerless posture estimation engine powered by artificial intelligence (AI) that can analyze human skeletal and posture information in both 2D and 3D. It can detect skeletal data for multiple individuals from still images and video. By creating training datasets and performing additional training, the accuracy of skeletal detection can be improved. This system does not require markers or depth sensors, and VisionPose can automatically identify up to twenty-five joints and five facial landmarks, totaling thirty anatomical locations [20]. This capability suggests that it could provide versatile and reliable assessments in clinical evaluation, as it has already demonstrated effectiveness in estimating knee joint posture in the frontal plane, showing potential applicability in clinical practice [21]. Therefore, with appropriate adaptation, VisionPose could become an accessible, affordable tool for widespread clinical use, providing effective motion analysis and potentially transformative treatment options for patients.

There is also a system similar to VisionPose called OpenPose, and research is also progressing in that area. Previous studies have compared the Vicon system with OpenPose across different motion contexts, including squatting [1] and treadmill walking [5], demonstrating high reliability and validity. Recent studies have further shown that joint position tracking during walking yields minimal discrepancies in joint position measurements between the two systems [22,23]. Specifically, studies investigating the validity of OpenPose’s 2D gait analysis for tasks involving 3D gait recognition across various ranges of motion (ROM) reported a significant correlation between the hip and knee joint angles measured by OpenPose and those obtained using Vicon’s 3D system [24]. Despite these findings, the reliability and validity of OpenPose for level ground walking assessments, especially using fixed cameras [5], remain underexplored. One study examined the femorotibial angle (FTA), the angle between the long axes of the femur and tibia, using VisionPose and a 3D optical motion analysis device. The results revealed a significant correlation between the FTA measurements obtained by the two systems, indicating the high validity of VisionPose as a tool for estimating knee joint posture in the frontal plane [21]. However, because these systems can only output the joint coordinates in a 2D image in pixel units [pix], previous studies did not mention important walking parameters such as stride length or walking speed, and the joint angles were simply calculated from the output coordinates [pix] obtained from an image showing the subject’s sagittal plane. In addition, the joint angle was not calculated from 3D coordinates, so there was a large error. In response to this, Imasen developed two types of systems (monocular camera type and compound camera type) that install cameras on the front and side of the subject, use VisionPose on the obtained images, and output parameters necessary for gait analysis such as 3D coordinates [mm] and joint angles from the output coordinates [pix]. The difference between the two developed systems is the angle of view of the camera installed on the side of the subject, and the two are used depending on the size of the measurement space. The reason for adopting VisionPose instead of OpenPose is due to the cost benefits when used commercially.

This study aimed to assess the reliability and validity of the Imasen system for gait analysis on level ground. Specifically, it evaluated monocular and composite camera configurations, referencing the Vicon system (the gold standard), to measure hip and knee joint angles, as well as time and distance parameters under various gait conditions, such as maximum speed and tandem walking. This study examined whether the Imasen system could be useful for both clinical and research purposes in gait analysis. Additionally, it hypothesized that the Imasen system would demonstrate comparable reliability to the Vicon system, accurately analyzing hip and knee joint movements during gait patterns like maximum speed and tandem walking. Furthermore, it anticipated that the Imasen system could achieve similar accuracy to traditional motion capture systems, while remaining simple and cost-effective.

## 2. Materials and Methods

### 2.1. Participants

Previous studies on internal validity with healthy individuals have shown that a minimum of 20 participants is typically required for reliable results [6,16,25]. Therefore, we included 11 participants in the monocular camera group (10 males and 1 females; mean age: 44.3 ± 10.3 years, mean height: 166.1 ± 6.1 cm, mean weight: 63.7 ± 10.3 kg) and 12 participants in the compound camera group (10 males and 2 females, mean age: 45.2 ± 11.3 years, mean height: 167.0 ± 6.9 cm, mean weight: 67.4 ± 14.3 kg), totaling 23 healthy adults. The subjects wore tight leggings and a shirt for all trials. Individuals with a history of severe injuries, such as ligament or musculoskeletal injuries, nerve damage, or fractures, were excluded from this study. The study protocol was approved by the Juntendo University Ethics Committee (#23-080), and all participants provided informed consent. Although there was a large gender difference in the samples this time, it is not thought that this would have a significant impact on the evaluation results.

The reason for this is that when Vicon affixes the markers, it is unavoidable that the measurement values will vary significantly depending on the skill level of the person applying the markers and the participant’s obesity level. However, it was thought that the difference in detection levels caused by differences in physique between men and women would be far smaller than this amount of change, and therefore the evaluation was based on the gender ratio this time.

### 2.2. Motion Task

Participants were assigned to walk on flat ground in the following order: comfortable, maximum speed, and tandem walking (walking with the toes of one foot in contact with the heel of the other foot) trials. At least 12 trials of three walking cycles each were video-recorded. No rest periods were provided, as all subjects were healthy and walking back and forth along the walking path was unlikely to cause fatigue.

### 2.3. Data Collection and Processing

A baseline internal parametric analysis was conducted using an eight-camera 3D Vicon configuration (Vantage Vero2.2 Camera, 2.2 MP resolution, 220 megapixels) (Figure 1).

Motion analysis was conducted using a Vicon motion capture system with a Plug-in Gait full-body model. A total of 39 reflective markers were placed on anatomical landmarks according to the standard Plug-in Gait protocol. These markers were used to capture the kinematics of the entire body, providing detailed three-dimensional motion data for subsequent analysis. The data were collected at a sampling frequency of [insert frequency, e.g., 100 Hz], ensuring high temporal resolution. Marker placement and data acquisition were performed by trained professionals to ensure accuracy and consistency throughout this study.

The Imasen Gait Evaluation System utilizes two types of cameras: a monocular camera and a composite camera. While this evaluation system employs a unique method for capturing gait data with cameras, the skeletal estimation process using VisionPose remains consistent. Gait data were collected using both the Imasen Gait Evaluation System and Vicon during the motion tasks, despite the differences in their systems.

A 2D Imasen camera configuration—a side camera (to record sagittal plane) and a front camera positioned orthogonal to the side camera (to record frontal plane)—was also used to simultaneously record the trials. The paired camera systems were prepared with different angles of view of the side cameras, and trials were recorded with each. With respect to the side cameras of the pair of camera systems, the camera with the low angle of view was a single camera (Figure 2), while the high one was a composite camera (to combine the images of four cameras placed close together to create a single image) (Figure 3). In the recording of a trial using the system in Figure 2, when the side camera was recording the right sagittal plane, the front camera was recording the frontal plane of the front; when the side camera was recording the left sagittal plane, the front camera was recording the frontal plane of the back. Both systems were calibrated using dedicated calibration software and a calibration board, and the evaluation was performed with the camera posture adjusted so that the images were vertical, horizontal, and parallel to the walking path.

The VisionPose algorithm was then used to process the data obtained from the Imasen cameras (version of VisionPose: V1.8.1).

### 2.4. Data Analysis

Peak angles of hip flexion and extension, as well as knee flexion and extension, were measured, including the ROM for each. Heel contact was identified via software analysis of frame-by-frame images to establish the gait cycle.

A total of 39 reflective markers were placed on anatomical landmarks following Vicon protocols. These markers were attached to the anterior and posterior superior iliac spines (ASIS and PSIS), the outer thigh, the lower leg, the lateral malleolus, the metatarsal heads, and the calcaneus. The coordinate system for the pelvic segment was defined using the ASIS and PSIS markers, with the origin set at the midpoint of the ASIS markers. The y-axis was aligned in the left–right direction, and the x-axis extended forward from the midpoint of the PSIS markers to the midpoint of the ASIS markers. This coordinate system structure was consistently applied to all segments.

Vicon software(NEXUS 2.0) was used to calculate pelvic angles and relative angles between coordinate systems using Euler angles. The pelvic ascent and descent angles were measured between the transverse axis (horizontal axis in the frontal plane) and the pelvic y-axis, reflecting the motions of the hip, knee, and ankle joints.

With respect to VisionPose, segment and joint angles were measured based on the estimated feature points of each joint. These angles were calculated using the coordinates of the reflective markers (Figure 4, Figure 5, Figure 6 and Figure 7).

The range of motion (ROM) for each gait cycle was determined by measuring gait parameters and the peak extension and flexion angles of the hip and knee joints. The starting point of the gait cycle was defined as the moment when the ankle joint center of the observed limb passed the hip joint center in the sagittal plane during the swing phase. To ensure the accurate identification of the same gait cycle, comparative validation was performed using the position of the floor reaction force meter linked to the Vicon system.

The calculation method for each gait parameter and joint angle during one trial with the Imasen system is outlined as follows:Record sagittal plane images with a side camera (lens distortion in images captured by the side camera with low angle of view was corrected. Lens distortion in images captured by the side composite cameras with high angle of view was corrected, and images from the four cameras were combined).Skeletal estimation using VisionPose was performed on images from both the side and front cameras, recording the coordinates of each joint node.Joint node coordinate information in each frame of the front camera was converted to real space coordinates, considering height information and joint node coordinates.Joint node coordinate information in each frame of the side camera was converted to real space coordinates, accounting for the distance between the side camera and walking path center plane.Each joint angle for each frame were calculated considering the angles between specific joint nodes in the sagittal plane.The gait cycle was identified based on the timing of the ankle joint center passing directly under the hip joint center.Each gait parameter, including stride time, stride length, and gait speed, was calculated.Peak angles and ROMs for each joint during the identified gait cycle were calculated.

Processes (1) and (2) were executed using a camera control program written in C++, while processes (3) through (8) were automatically handled by a dedicated Python program. Additionally, IBM SPSS v.24.0 (IBM, Tokyo, Japan) was used to compare and validate the outputs of the Vicon and Imasen systems.

In the image processing workflow (Figure 7), lens distortion correction and projection transformation were carried out using OpenCV functions. For the combined camera with a wide field of view, images were projectively transformed relative to the 1.34 m projection plane (using OpenCV’s warpPerspective function), and the four transformed images were merged into a sagittal plane image (using OpenCV’s matchTemplate function). For the side camera with a narrow field of view, only lens distortion correction was performed as it involved a single camera. These adjustments were finalized by mounting the camera on a highly rigid pedestal, eliminating the need for recalibration during system relocation or reinstallation.

### 2.5. Statistical Analysis

ICCs and 95% confidence intervals (CIs) were used to assess test–retest, intra-rater, and inter-rater reliabilities. To evaluate the agreement between the Imasen system and Vicon, ICCs were calculated, with ICCs < 0.5 indicating poor agreement, 0.5–0.75 indicating moderate agreement, 0.75–0.9 indicating good agreement, and >0.9 indicating excellent agreement [23].

Linear regression was used to analyze the correlation between the Imasen system and Vicon data. Linear regression was used to analyze correlations between the Imasen system and Vicon data, with correlation coefficients (*r*) as follows: 0.1–0.3 (small correlation), 0.3–0.5 (medium correlation), and >0.5 (large correlation) [24]. The correlation coefficient (*r*) was interpreted as follows: 0.2–0.4 (small correlation), 0.4–0.7 (medium correlation), and values greater than 0.7 (large correlation). The coefficient of determination (R^2^) was interpreted as follows: 0.04–0.16 (small correlation), 0.16–0.5 (medium correlation), and values greater than 0.5 (large correlation). For Vicon vs. Imasen, internal consistency was assessed using Cronbach’s alpha, where values ≥ 0.7 were deemed acceptable [25,26,27]. All statistical analyses were performed using IBM SPSS v.24.0 (IBM, Tokyo, Japan), with a significance level set at *p* < 0.05.

## 3. Results

### 3.1. Data Acquisition Rate

In this study, the data acquisition success rates were 97.3% for the Vicon system, 100.0% for the monocular camera, and 97.5% for the composite camera (Table 1a,b). However, none of the systems were able to consistently and accurately analyze all aspects of gait. The Vicon system faced challenges in capturing certain movements due to inconsistencies in calculated cadence values, missed markers during gait cycles, and errors in gait parameter output. Nonetheless, successful analyses were documented.

### 3.2. Test–Retest and Intra-Rater Reliability Analysis

Figure 8 and Figure 9 display hip and knee joint angle changes during one gait cycle across three gait patterns, as recorded by the Vicon and monocular camera systems. Figure 10 and Figure 11 show similar data using the Vicon and composite camera systems. Key points include:Summary of joint angle changes across three gait patterns.Gait cycle normalization, where 0% represents the point at which the hip joint’s central axis crosses over the ankle joint’s central axis, with 100% representing one complete gait cycle.Solid lines represent the mean values for monocular and composite camera systems, while dashed lines denote Vicon system values, with standard deviation (SD) values also provided.

The data calculated separately by Vicon and the Imasen Gait Evaluation System were compared. The differences in hip joint flexion angles between Vicon and the Imasen Gait Evaluation System were 7.9° during comfortable walking, 9.1° during maximum walking, and 4.8° during tandem walking (monocular), and 7.5°, 8.0°, and 5.4° (composite), respectively, with the composite method showing smaller errors. For hip joint extension angles, the differences were −7.1° during comfortable walking, −6.2° during maximum walking, and −7.2° during tandem walking (monocular), and −5.3°, −4.5°, and −6.6° (composite), respectively, with the composite method, again showing smaller errors. In knee joint flexion angles, the differences were 9.5° during comfortable walking, 7.7° during maximum walking, and 6.4° during tandem walking (monocular), and 8.8°, 7.8°, and 7.2° (composite), respectively, with similar results for both the monocular and composite methods. For knee joint extension angles, the differences during comfortable walking were 7.3°, 6.4°, and 6.0° for the monocular method, and 5.0°, 2.6°, and 4.2° for the composite method, with the composite method demonstrating smaller errors.

The changes in joint angles for each of the three gait patterns are summarized. The identification of one gait cycle was normalized, so that zero was the time when the central axis of the hip joint crossed over the central axis of the ankle joint, and 100% of one gait cycle was identified. The solid line indicates the mean value for all subjects in the monocular camera gait system, and the dashed line indicates the mean value for all subjects in the Vicon system. The SD is also described.

Table 2a and Table 3a show the comparison results between the monocular camera, Table 2b and Table 3b show the comparison results between the combined camera and the Vicon system, with ICC (intraclass correlation coefficient), mean and standard deviation (SD) of the gait parameters for the three ground walking sessions. parameters exceeded 0.936, and monocular and combined cameras showed similarly high ICC values. Test-retest reliability was very high for all items.

### 3.3. Criterion Validity

The ICC for time–distance gait parameters was very high for the monocular camera (ICC ≥ 0.933, *p* < 0.001, Table 1a) and good to very high for the composite camera (ICC ≥ 0.837, *p* < 0.05, Table 1b). Regarding ROM, except for the hip extension angle during tandem walking with the composite camera (ICC: 0.314, *p* = 0.571), results were good to very high across all conditions for both the monocular camera (ICC ≥ 0.826, *p* < 0.02, Table 4a) and the composite camera (ICC ≥ 0.86, *p* < 0.005, Table 4b).

## 4. Discussion

The ICC(1,k) indicated strong test–retest reliability between the markerless and marker-based systems. ICC(2,k) and ICC(3,k) demonstrated high reliability for time and distance parameters, confirming minimal differences between the devices. Measurements showed high reliability and validity, with low error rates. Most hip flexion-extension ROM and knee joint parameters exhibited high coefficients of determination (R^2^) without proportional biases, supporting the accuracy of the Imasen system. However, the hip extension angle during tandem walking had significantly reduced reliability (monocular and composite cameras), likely due to the Imasen system’s limitation as a 2D analysis system, which struggled with transverse plane rotations of the knee, pelvis, and trunk. For tasks not involving such rotations, results remained valid. Differences in hip joint angle measurements between Vicon and the Imasen system were attributed to variations in pelvis and spine movement tracking. Overall, the Imasen system showed good to excellent ICCs for most lower-limb ROM parameters, with narrow 95% CIs.

There were high coefficients of determination between the data obtained by Vicon and the Imasen system for the hip flexion-extension ROM and most knee parameters. Furthermore, no proportional biases were observed. The ICCs for most ROM parameters in the sagittal plane of the lower extremity ranged from good to excellent, with a narrow 95% CI. Therefore, the validity of the Imasen system for lower-limb ROM is supported. However, the reliability of the hip extension angle during tandem walking was significantly reduced when using monocular and composite camera systems. This indicates that movements involving the lateral rotation of the knee and the rotation of the pelvis and trunk cannot be measured using monocular and composite camera systems. This is because the Imasen system is inherently a 2D image analysis system, making it unable to accurately track transverse plane rotations across multiple frames. However, valid results were obtained for tasks that did not involve transverse plane rotations. Additionally, differences in the methods of measuring hip joint angles were observed between Vicon and the Imasen system due to the movements of the pelvis and spine.

Several previous studies have analyzed gait and other movement patterns using markerless analysis [1,23]. The minimum detectable change in temporal gait parameters obtained during intersession and test–retest experiments on healthy human gait using 3D motion capture has been reported to range from 0.02 to 0.08 s [28,29]. The results of this study are consistent with previous reports on 3D markerless gait evaluation using a fixed camera, and the Imasen system provides promising quantitative information. This system is expected to enable low-cost, simplified motion analysis. Further verification of the reliability and validity of the device should be a focus of future research. Furthermore, the system requires only a small number of cameras, and its accuracy has been widely verified [30]. Typically, the number of cameras and other factors also affect measurement accuracy, making it suitable for clinical use but somewhat limited for research purposes. Skeletal estimation has demonstrated very high accuracy and can be used clinically. However, it is important to understand that Vicon and the Imasen system have different joint angle settings. Additionally, using these images for clinical evaluation (e.g., the Edinburgh Visual Gait Score) [31] may reduce the therapist’s evaluation burden. Simplified movement acquisition would also allow for data accumulation from many individuals. A recent study showed that teaching running to students using a feature extraction algorithm to analyze movement features from KinectV2 significantly improved their running speed [32]. The development of systems that facilitate the easy capture of routine clinical data using markerless analysis, extract this data using feature extraction algorithms or other methods, and provide feedback to patients is expected to aid in planning treatment programs. This study has several limitations. First, the monocular and composite camera setups used here were limited to detecting movements within the sagittal plane, specifically tracking hip and knee joints.

This study has another limitation. The study sample was restricted to healthy adults, which limited the generalizability of the findings regarding the reliability and validity of the 2D systems for 3D tasks. However, this study also demonstrated the system’s capacity to analyze various gait types on flat ground, rather than the treadmill-based analysis, which was significant for future research.

## 5. Conclusions

This study confirms the effectiveness of the Imasen system AI engine for posture estimation in 3D gait analysis. Specifically, the Imasen system showed significant correlation with Vicon’s 3D measurements for hip and knee flexion-extension angles, presenting only a fixed bias but no proportional bias, and displaying high coefficients of determination and robust ICCs for most lower extremity sagittal plane parameters. While gait speed variations did not significantly impact time–distance ICCs, certain complex movements such as hip joint angles during tandem walking revealed reduced reliability and validity. This suggests that the Imasen system may be less effective than 3D motion analysis systems. Nevertheless, the Imasen system remains a promising alternative to traditional 3D motion analysis systems and could potentially be utilized for specific gait recognition tasks. Future research will evaluate the potential cost and time savings of the Imasen system, as well as its robustness to include people with body shape and skeletal characteristics such as hunched backs and obesity. The huge benefits it brings to society need to be thoroughly investigated.

## Figures and Tables

**Figure 1 sensors-25-01076-f001:**
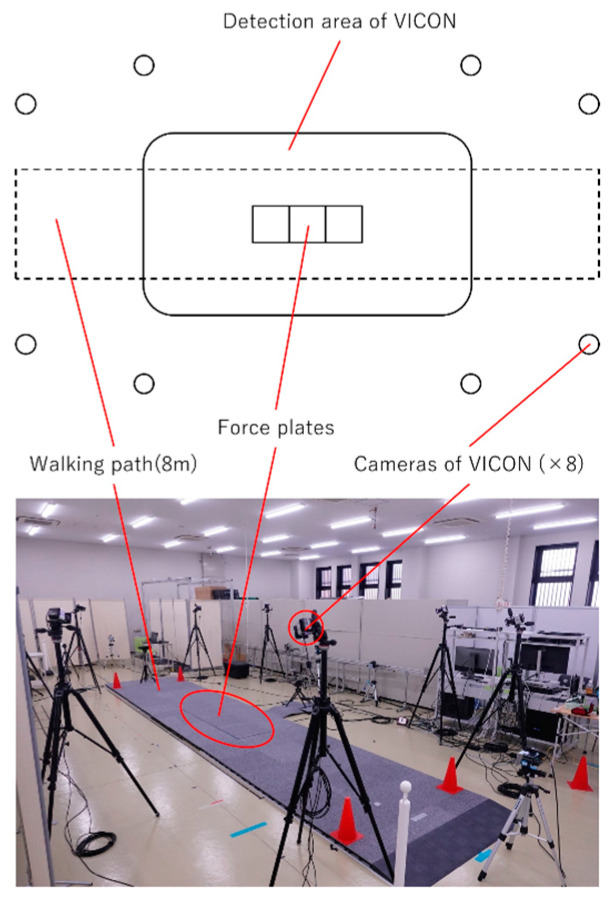
Vicon gait measurement environment.

**Figure 2 sensors-25-01076-f002:**
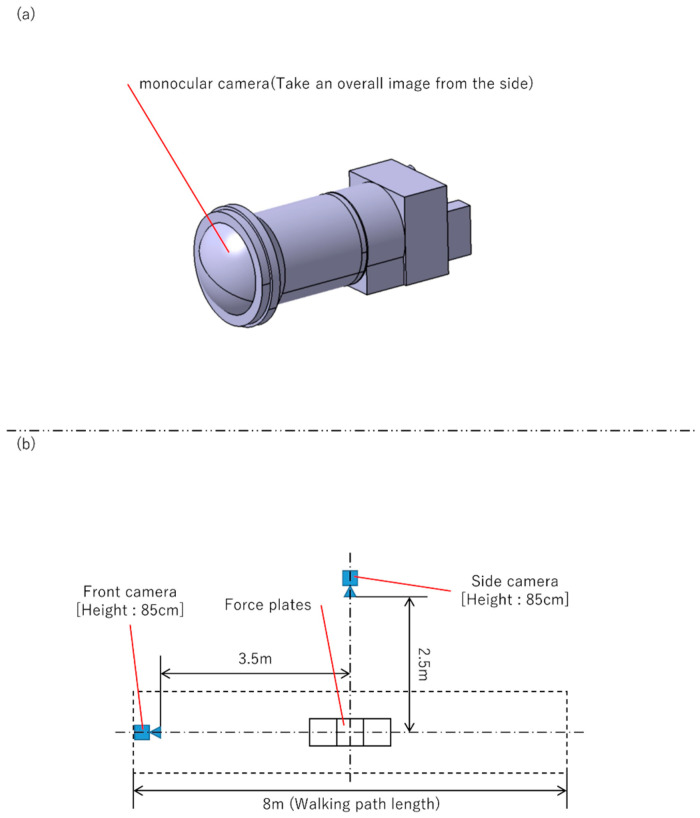
Monocular camera and VisionPose measurement environment, featuring a side camera with a low angle of view. (**a**) Structure of the monocular camera. (**b**) Layout of the walking paths and camera placements.

**Figure 3 sensors-25-01076-f003:**
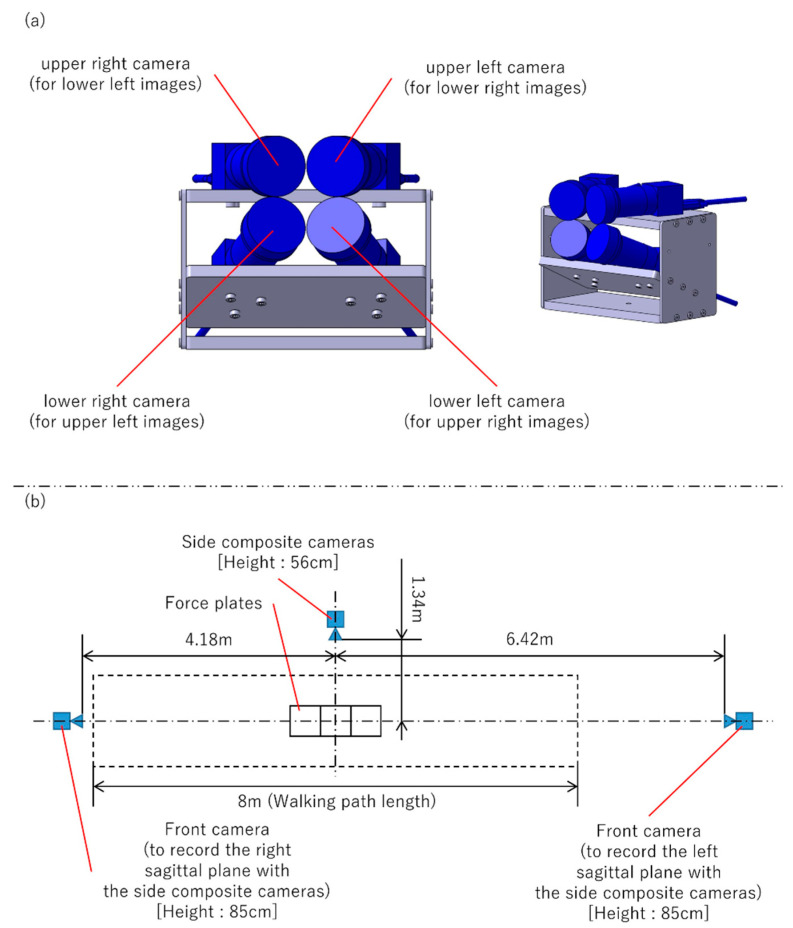
Composite camera and VisionPose measurement environment, with side composite cameras at a high angle of view. (**a**) Side composite camera structure. (**b**) Layout of the walking paths and camera placements.

**Figure 4 sensors-25-01076-f004:**
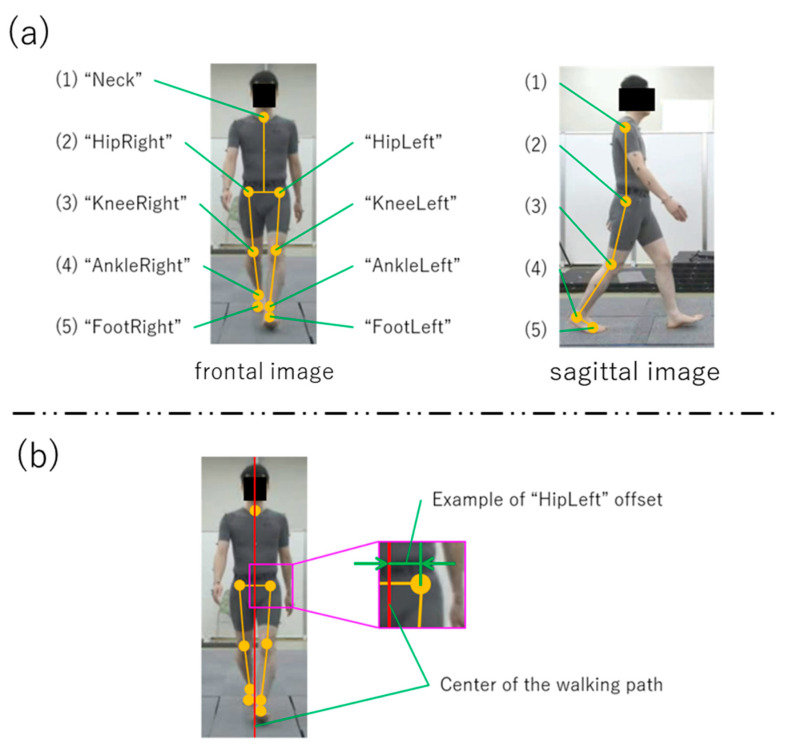
VisionPose nodes. (**a**) Nodes at each skeletal estimate location. (**b**) Center of walking path and offset. Each coordinate from the front camera was converted to real-world coordinates, with the offset calculated from the center of the walking path.

**Figure 5 sensors-25-01076-f005:**
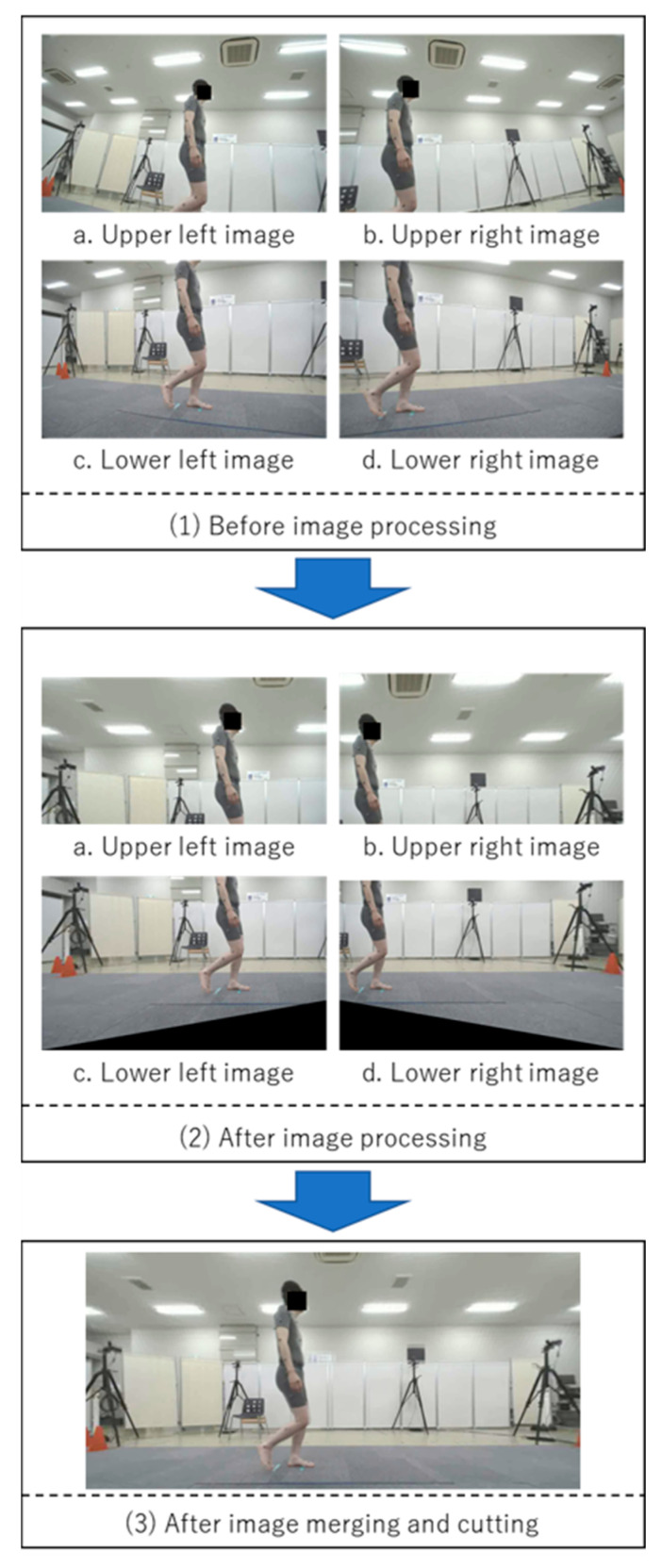
Processing flow for combining images from the side composite camera.

**Figure 6 sensors-25-01076-f006:**
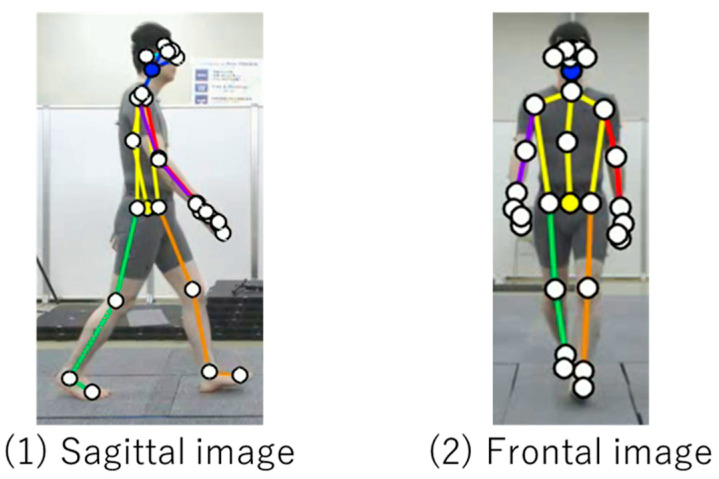
Example of Imasen system application.

**Figure 7 sensors-25-01076-f007:**
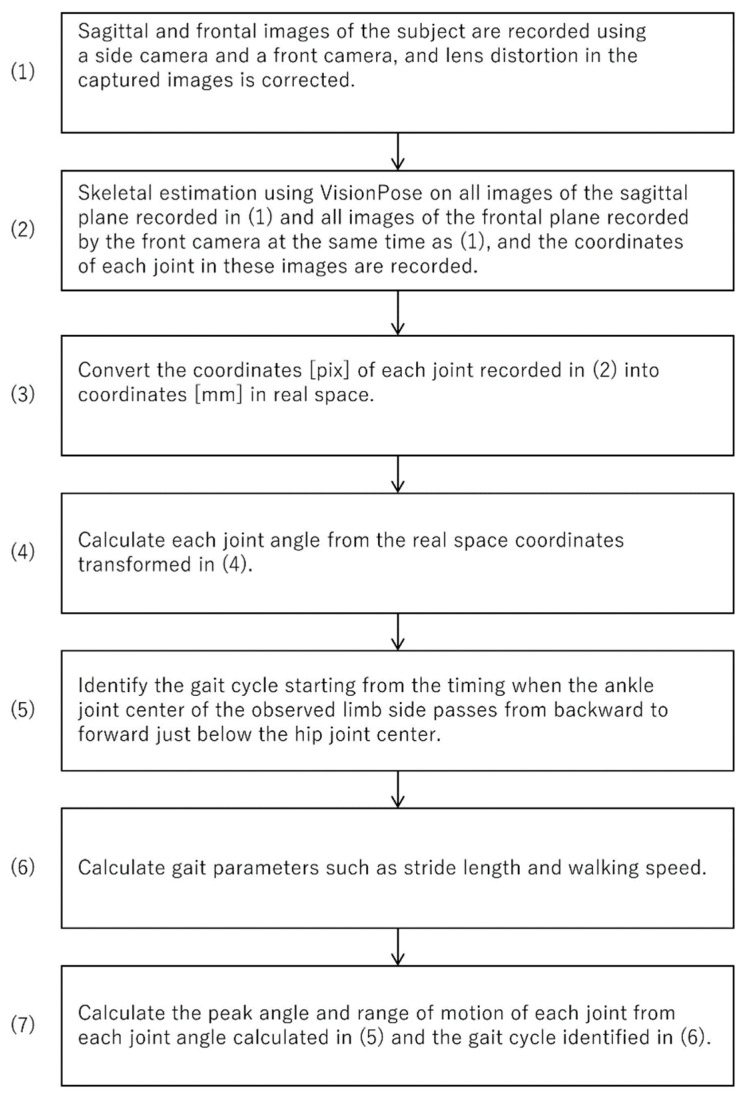
Imasen system walk and gait parameter outputs.

**Figure 8 sensors-25-01076-f008:**
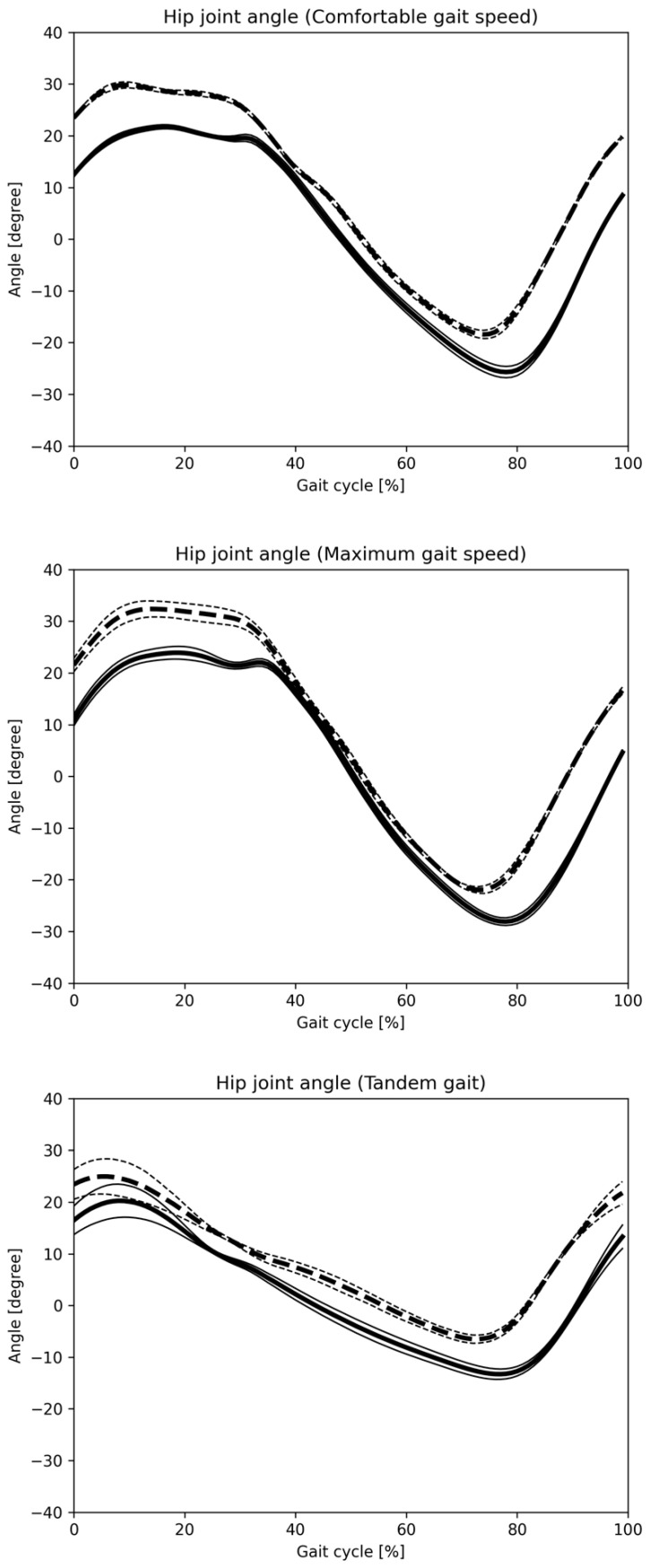
Hip joint angle change in one gait cycle (monocular camera). The changes in joint angles for each of the three gait patterns are summarized. The identification of one gait cycle was normalized, so that 0 was the time when the central axis of the hip joint crossed over the central axis of the ankle joint, and 100% of one gait cycle was identified. The solid line indicates the mean value for all subjects in the Imasen gait system, and the dashed line indicates the mean value for all subjects in the VICON system. The SD is also described.

**Figure 9 sensors-25-01076-f009:**
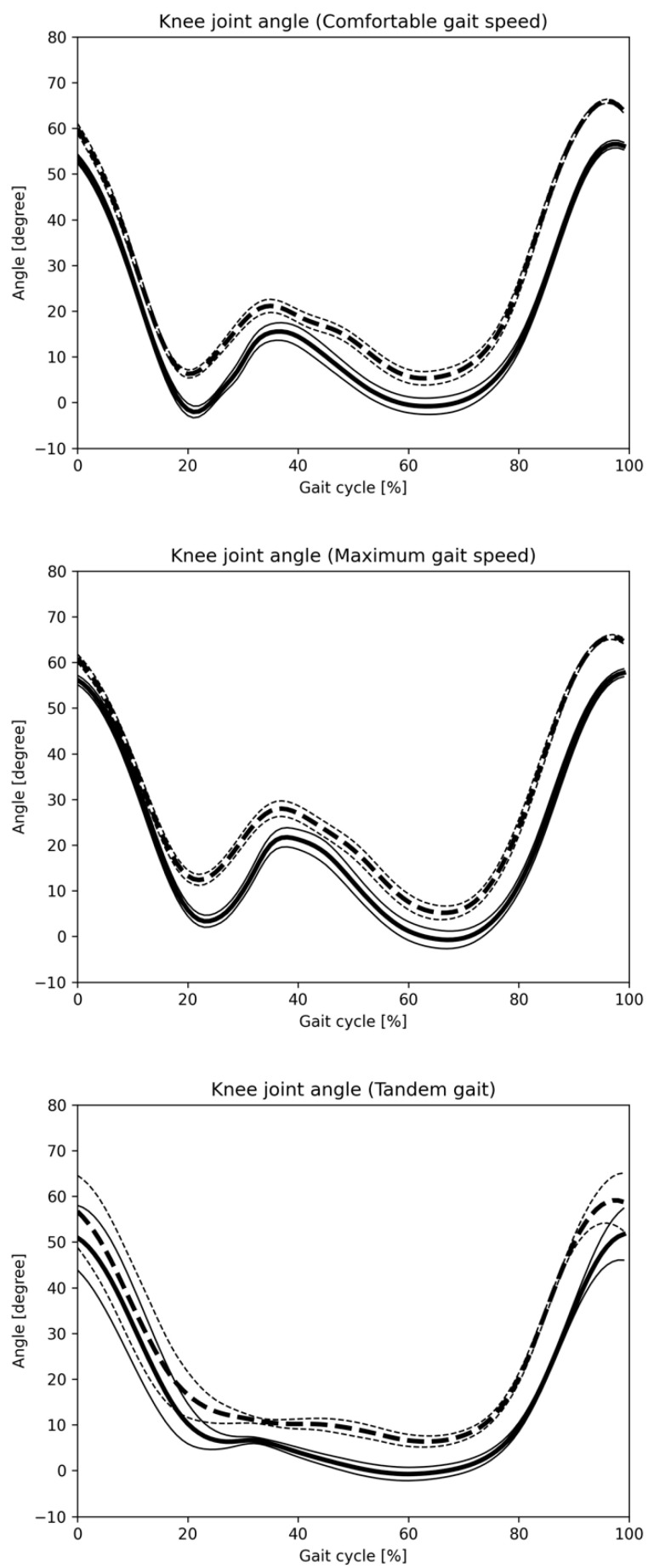
Knee joint angle change in one gait cycle (monocular camera). The changes in joint angles for each of the three gait patterns are summarized. The identification of one gait cycle was normalized, so that 0 was the time when the central axis of the hip joint crossed over the central axis of the ankle joint, and 100% of one gait cycle was identified. The solid line indicates the mean value for all subjects in the Imasen gait system, and the dashed line indicates the mean value for all subjects in the VICON system. The SD is also described.

**Figure 10 sensors-25-01076-f010:**
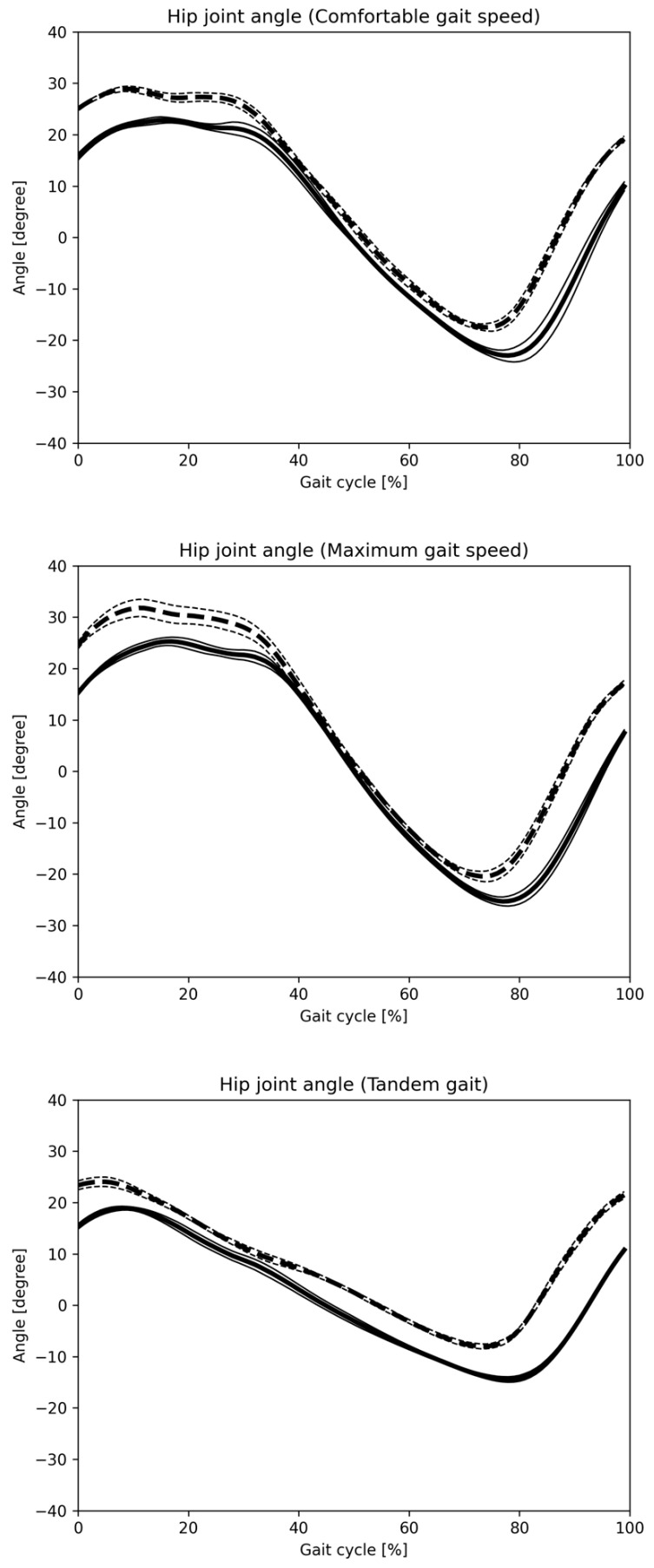
Hip joint angle change in one gait cycle (composite camera). The changes in joint angles for each of the three gait patterns are summarized. The identification of one gait cycle was normalized, so that 0 was the time when the central axis of the hip joint crossed over the central axis of the ankle joint, and 100% of one gait cycle was identified. The solid line indicates the mean value for all subjects in the Imasen gait system, and the dashed line indicates the mean value for all subjects in the VICON system. The SD is also described.

**Figure 11 sensors-25-01076-f011:**
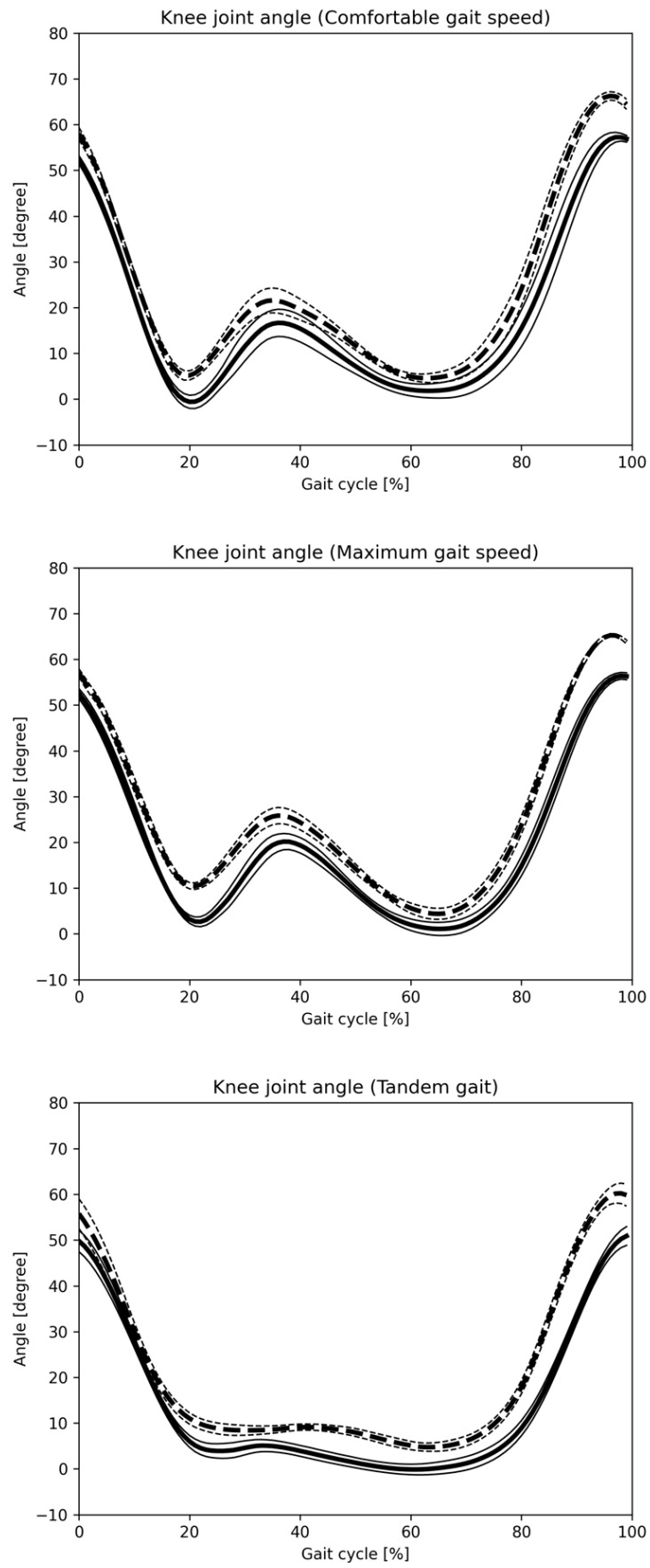
Knee joint angle change in one gait cycle (composite camera). The changes in joint angles for each of the three gait patterns are summarized. The identification of one gait cycle was normalized, so that 0 was the time when the central axis of the hip joint crossed over the central axis of the ankle joint, and 100% of one gait cycle was identified. The solid line indicates the mean value for all subjects in the Imasen gait system, and the dashed line indicates the mean value for all subjects in the VICON system. The SD is also described.

**Table 1 sensors-25-01076-t001:** (**a**) The regression models and the intraclass correlation coefficients [ICC(2,k)] for gait item using Vicon and a monocular camera gait measurement system. (**b**) The regression models and the intraclass correlation coefficients [ICC(2,k)] for gait item using Vicon and a composite camera gait measurement system.

(a) ^1^							
Item	Gait Pattern	Unstandardized Coefficients B	Constant	95% CI for B (*p* Value)	R^2^	ICC(2,k)	Cronbach’s Alpha
Step length	Com	0.960	**50.510**	**0.918 to 1.002 (*p* < 0.001)**	** 0.996 **	** 0.983 **	0.999
	Max	0.999	**54.037**	**0.958 to 1.041 (*p* < 0.001)**	** 0.996 **	** 0.991 **	0.999
	Tandem	1.031	**17.321**	**0.898 to 1.163 (*p* < 0.001)**	** 0.965 **	** 0.990 **	0.991
Gait speed	Com	0.933	**36.376**	**0.876 to 0.990 (*p* < 0.001)**	** 0.992 **	** 0.981 **	0.997
	Max	0.971	**12.855**	**0.802 to 1.139 (*p* < 0.001)**	** 0.937 **	** 0.989 **	0.998
	Tandem	1.052	**39.057**	**0.992 to 1.112 (*p* < 0.001)**	** 0.993 **	** 0.997 **	0.998
Stride time	Com	1.039	**5.964**	**0.651 to 1.427 (*p* < 0.001)**	** 0.759 **	** 0.933 **	0.932
	Max	0.971	**12.855**	**0.802 to 1.139 (*p* < 0.001)**	** 0.937 **	** 0.986 **	0.985
	Tandem	1.042	**37.162**	**0.979 to 1.104 (*p* < 0.001)**	** 0.992 **	** 0.998 **	0.998
Stride length	Com	0.969	**40.839**	**0.916 to 1.021 (*p* < 0.001)**	** 0.993 **	** 0.986 **	0.998
	Max	1.008	**49.703**	**0.963 to 1.054 (*p* < 0.001)**	** 0.996 **	** 0.994 **	0.999
	Tandem	1.017	**25.353**	**0.927 to 1.106 (*p* < 0.001)**	** 0.983 **	** 0.988 **	0.995
**(b) ^2^**							
**Item**	**Gait Pattern**	**Unstandardized Coefficients B**	**Constant**	**95% CI for B (*p* Value)**	**R^2^**	**ICC(2,k)**	**Cronbach’s Alpha**
Step length	Com	0.939	**28.184**	**0.864 to 1.015 (*p* < 0.001)**	** 0.988 **	** 0.984 **	0.996
	Max	0.95	**19.974**	**0.842 to 1.057 (*p* < 0.001)**	** 0.975 **	** 0.981 **	0.994
	Tandem	0.837	**3.018**	**0.210 to 1.454 (*p* = 0.015)**	** 0.448 **	** 0.837 **	0.823
Gait speed	Com	0.989	**38.114**	**0.934 to 1.048 (*p* < 0.001)**	** 0.993 **	** 0.994 **	0.998
	Max	0.974	**31.132**	**0.903 to 1.045 (*p* < 0.001)**	** 0.990 **	** 0.994 **	0.998
	Tandem	0.905	**17.715**	**0.790 to 1.021 (*p* < 0.001)**	** 0.969 **	** 0.991 **	0.991
Stride time	Com	1.128	**11.166**	**0.899 to 1.356 (*p* < 0.001)**	** 0.925 **	** 0.899 **	0.977
	Max	0.936	**18.261**	**0.820 to 1.052 (*p* < 0.001)**	** 0.971 **	** 0.949 **	0.993
	Tandem	0.996	**55.667**	**0.955 to 1.036 (*p* < 0.001)**	** 0.997 **	** 0.995 **	0.999
Stride length	Com	0.985	**30.044**	**0.910 to 1.059 (*p* < 0.001)**	** 0.989 **	** 0.991 **	0.997
	Max	0.99	**25.562**	**0.902 to 1.077 (*p* < 0.001)**	** 0.985 **	** 0.991 **	0.997
	Tandem	0.917	**4.921**	**0.495 to 1.339 (*p* < 0.001)**	** 0.699 **	** 0.927 **	0.920

^1^ Abbreviations: Com, comfortable gait speed; Max, maximum gait speed; Tandem, tandem gait; ICC, intraclass correlation coefficient; CI, confidence intervals. The linear regression analyses were performed by using the data obtained by Monocular camera gait measurement system as the independent variable and the data obtained by Vicon as the dependent variable. Only statistically significant variables in regression analysis (*p* values < 0.05) are shown in bold. Moreover, the variables exceed thresholds (R^2^ > 0.25, ICC > 0.5) are underlined. ^2^ Abbreviations: Com, comfortable gait speed; Max, maximum gait speed; Tandem, tandem gait; ICC, intraclass correlation coefficient; CI, confidence intervals. The linear regression analyses were performed by using the data obtained by composite camera gait measurement system as the independent variable and the data obtained by Vicon as the dependent variable. Only statistically significant variables in regression analysis (*p* values < 0.05) are shown in bold. Moreover, the variables exceed thresholds (R^2^ > 0.25, ICC > 0.5) are underlined.

**Table 2 sensors-25-01076-t002:** (**a**) The mean ± standard deviations for time–distance factor using Vicon and a monocular camera gait measurement system. (**b**) The mean ± standard deviations for time–distance factor using Vicon and a composite camera gait measurement system.

(a) ^1^								
Measurement		Mean Data					Test-Retest ICC(1,k)	
		Vicon		Monocular Camera		Diff	Vicon	Monocular Camera
		Mean	SD	Mean	SD			
Step length (mm)	Com	694.8	74.0	714.4	76.9	19.6	0.997	0.997
	Max	799.7	102.6	818.9	102.5	19.2	0.997	0.997
	Tandem	328.8	32.0	332.0	30.5	3.2	0.986	0.985
Gait speed (m/s)	Com	1.33	0.16	1.37	0.17	0.04	0.996	0.996
	Max	1.87	0.30	1.93	0.31	0.06	0.997	0.997
	Tandem	0.58	0.14	0.59	0.13	0.01	0.997	0.996
Stride time (s)	Com	1.04	0.04	1.05	0.04	0.00	0.960	0.982
	Max	0.86	0.05	0.86	0.05	0.00	0.984	0.990
	Tandem	1.18	0.21	1.17	0.20	−0.01	0.994	0.997
Stride length (mm)	Com	1382.9	142.4	1416.2	146.6	33.3	0.998	0.998
	Max	1587.2	198.0	1617.6	195.9	30.5	0.998	0.999
	Tandem	658.0	63.2	670.2	61.7	12.2	0.990	0.990
**(b) ^2^**								
**Measurement**		**Mean Data**					**Test-Retest ICC(1,k)**	
		**Vicon**		**Composite Camera**		**Diff**	**Vicon**	**Composite Camera**
		**Mean**	**SD**	**Mean**	**SD**			
Step length (mm)	Com	724.8	104.2	750.3	110.3	25.5	0.994	0.995
	Max	833.4	116.3	862.7	121.1	29.3	0.997	0.997
	Tandem	371.5	54.0	379.5	54.8	8.0	0.984	0.978
Gait speed (m/s)	Com	1.37	0.23	1.40	0.23	0.03	0.996	0.996
	Max	1.88	0.32	1.92	0.33	0.04	0.998	0.998
	Tandem	0.62	0.16	0.64	0.17	0.02	0.995	0.995
Stride time (s)	Com	1.09	0.05	1.06	0.04	−0.03	0.953	0.982
	Max	0.91	0.05	0.89	0.06	−0.02	0.978	0.990
	Tandem	1.21	0.23	1.18	0.23	−0.03	0.991	0.996
Stride length (mm)	Com	1446.9	206.6	1481.9	208.8	35.0	0.998	0.997
	Max	1662.5	229.2	1699.9	230.0	37.4	0.999	0.998
	Tandem	737.3	102.8	749.2	105.6	11.9	0.989	0.987

^1^ This table shows the means, standard deviations and ICCs of the time factors from the Vicon and monocular camera instruments. Abbreviations: Com, comfortable gait speed; Max, maximum gait speed; Tandem, tandem gait; SD, standard deviation; Diff, difference; Monocular camera, monocular camera gait measurement system. ^2^ This table shows the means, standard deviations and ICCs of the time factors from the Vicon and composite camera instruments. Abbreviations: Com, comfortable gait speed; Max, maximum gait speed; Tandem, tandem gait; SD, standard deviation; Diff, difference; Composite camera, composite camera gait measurement system.

**Table 3 sensors-25-01076-t003:** (**a**) Mean values ± standard deviation (degree) for each peak angle and peak phase using Vicon and a monocular camera gait measure system. (**b**) Mean values ± standard deviation (degree) for each peak angle and peak phase using Vicon and a composite camera gait measurement system.

(a) ^1^									
Item			Mean Data					Test-Retest ICC(1,k)	
			Vicon		Monocular Camera		Diff	Vicon	Monocular Camera
			Mean	SD	Mean	SD			
Hip joint	Flexion angle (°)	Com	22.5	1.9	30.3	2.3	7.9	0.976	0.983
		Max	26.3	3.8	35.4	5.5	9.1	0.991	0.995
		Tandem	20.3	4.8	25.2	4.7	4.8	0.990	0.994
	Extension angle (°)	Com	25.9	2.3	18.8	2.1	−7.12	0.985	0.980
		Max	28.8	2.2	22.6	2.7	−6.23	0.981	0.985
		Tandem	13.4	2.1	6.1	1.8	−7.24	0.936	0.969
Knee joint	Flexion angle (°)	Com	56.2	4.1	65.7	2.5	9.53	0.974	0.976
		Max	58.2	5.0	65.9	2.9	7.68	0.979	0.976
		Tandem	54.3	10.2	60.8	9.3	6.43	0.994	0.995
	Extension angle (°)	Com	−4.8	3.2	2.5	3.2	7.3	0.987	0.991
		Max	−3.1	3.7	3.3	3.9	6.4	0.987	0.988
		Tandem	−1.2	3.4	4.9	2.6	6.0	0.953	0.969
**(b) ^2^**									
**Item**			**Mean Data**					**Test-Retest ICC(1,k)**	
			**VICON**		**Composite Camera**		**Diff**	**Vicon**	**Composite Camera**
			**Mean**	**SD**	**Mean**	**SD**			
Hip joint	Flexion angle (°)	Com	24.0	2.6	31.5	3.7	7.5	0.981	0.983
		Max	28.3	3.6	36.3	4.8	8.0	0.979	0.988
		Tandem	20.0	1.7	25.5	2.4	5.4	0.948	0.966
	Extension angle (°)	Com	25.4	3.0	20.1	2.7	−5.3	0.980	0.971
		Max	28.8	3.4	24.2	3.4	−4.5	0.978	0.984
		Tandem	15.5	2.5	8.9	1.8	−6.6	0.941	0.963
Knee joint	Flexion angle (°)	Com	56.7	3.7	65.5	3.4	8.8	0.975	0.968
		Max	58.7	3.6	66.5	2.7	7.8	0.967	0.988
		Tandem	52.8	5.8	60.0	6.2	7.2	0.986	0.978
	Extension angle (°)	Com	−5.6	3.1	−0.5	3.7	5.0	0.986	0.973
		Max	−3.7	3.2	−1.1	4.0	2.6	0.982	0.984
		Tandem	−4.4	4.5	−0.2	3.7	4.2	0.968	0.974

^1^ This table shows the means, standard deviations and ICCs of the time factors from the Vicon and monocular camera instruments. Abbreviations: Com, comfortable gait speed; Max, maximum gait speed; Tandem, tandem gait; SD, standard deviation; Diff, difference; Monocular camera, monocular camera gait measurement system. ^2^ This table shows the means, standard deviations and ICCs of the time factors from the Vicon and composite camera instruments. Abbreviations: Com, comfortable gait speed; Max, maximum gait speed; Tandem, tandem gait; SD, standard deviation; Diff, difference; Composite camera, composite camera gait measurement system.

**Table 4 sensors-25-01076-t004:** (**a**) The regression models and the intraclass correlation coefficients [ICC(3,k)] for each peak angle and peak phase using Vicon and a monocular camera gait. (**b**) The regression models and the intraclass correlation coefficients [ICC(3,k)] for each peak angle and peak phase using Vicon and a composite camera gait measurement system.

(a) ^1^							
Item	Gait Pattern	Unstandardized Coefficients B	Constant	95% CI for B (*p* Value)	R^2^	ICC(3,k)	Cronbach’s Alpha
Hip							
Com	Flexion	0.695	**4.363**	**0.340 to 1.050 (*p* = 0.001)**	** 0.621 **	** 0.889 **	0.889
	Extension	1.045	**8.126**	**0.759 to 1.333 (*p* < 0.001)**	** 0.855 **	** 0.961 **	0.961
Max	Flexion	0.652	**8.634**	**0.484 to 0.820 (*p* < 0.001)**	** 0.870 **	** 0.936 **	0.936
	Extension	0.734	**5.913**	**0.457 to 1.010 (*p* < 0.001)**	** 0.755 **	** 0.929 **	0.929
Tandem	Flexion	0.979	**10.558**	**0.772 to 1.185 (*p* < 0.001)**	** 0.909 **	** 0.978 **	0.978
	Extension	0.800	**3.175**	**0.238 to 1.361 (*p* = 0.010)**	** 0.452 **	** 0.826 **	0.826
Knee							
Com	Flexion	1.405	**5.776**	**0.863 to 1.947 (*p* < 0.001)**	** 0.746 **	** 0.881 **	0.881
	Extension	0.806	**4.110**	**0.369 to 1.242 (*p* = 0.002)**	** 0.806 **	** 0.884 **	0.884
Max	Flexion	1.499	**5.933**	**0.936 to 2.062 (*p* < 0.001)**	** 0.757 **	** 0.871 **	0.871
	Extension	0.814	**5.393**	**0.478 to 1.151 (*p* < 0.001)**	** 0.719 **	** 0.925 **	0.925
Tandem	Flexion	1.063	**13.724**	**0.890 to 1.235 (*p* < 0.001)**	** 0.945 **	** 0.985 **	0.985
	Extension	1.184	**8.003**	**0.854 to 1.514 (*p* < 0.001)**	** 0.851 **	** 0.949 **	0.949
**(b) ^2^**							
**Item**	**Gait Pattern**	**Unstandardized Coefficients B**	**Constant**	**95% CI for B (*p* Value)**	**R^2^**	**ICC(3,k)**	**Cronbach’s Alpha**
Hip							
Com	Flexion	0.628	**6.567**	**0.412 to 0.845 (*p* < 0.001)**	** 0.808 **	** 0.919 **	0.919
	Extension	0.943	**4.772**	**0.496 to 1.390 (*p* = 0.001)**	** 0.685 **	** 0.914 **	0.914
Max	Flexion	0.688	**7.631**	**0.484 to 0.892 (*p* < 0.001)**	** 0.851 **	** 0.942 **	0.942
	Extension	0.856	**4.850**	**0.457 to 1.256 (*p* < 0.001)**	** 0.692 **	** 0.919 **	0.919
Tandem	Flexion	0.625	**4.687**	**0.323 to 0.926 (*p* = 0.001)**	** 0.677 **	** 0.893 **	0.893
	Extension	0.250	0.589	−0.710 to 1.210 (*p* = 0.571)	−0.070	0.314	0.314
Knee							
Com	Flexion	1.040	**9.369**	**0.789 to 1.292 (*p* < 0.001)**	** 0.897 **	** 0.974 **	0.974
	Extension	0.611	**3.316**	**0.194 to 1.028 (*p* = 0.009)**	** 0.500 **	** 0.925 **	0.925
Max	Flexion	1.070	**4.275**	**0.504 to 1.636 (*p* < 0.002)**	** 0.633 **	** 0.883 **	0.883
	Extension	0.623	**3.637**	**0.235 to 1.010 (*p* = 0.005)**	** 0.550 **	** 0.86 **	0.86
Tandem	Flexion	0.880	**8.033**	**0.632 to 1.128 (*p* < 0.001)**	** 0.864 **	** 0.966 **	0.966
	Extension	0.757	**2.589**	**0.096 to 1.418 (*p* = 0.029)**	** 0.363 **	** 0.785 **	0.785

^1^ Abbreviations: Com, comfortable gait speed; Max, maximum gait speed; Tandem, tandem gait; ICC, intraclass correlation coefficient; CI, confidence intervals. The linear regression analyses were performed by using the data obtained by monocular camera gait measurement system as the independent variable and the data obtained by Vicon as the dependent variable. Only statistically significant variables in regression analysis (*p* values < 0.05) are shown in bold. Moreover, the variables exceed thresholds (R^2^ > 0.25, ICC > 0.5) are underlined. ^2^ Abbreviations: Com, comfortable gait speed; Max, maximum gait speed; Tandem, tandem gait; ICC, intraclass correlation coefficient; CI, confidence intervals. The linear regression analyses were performed by using the data obtained by a composite camera gait measurement system as the independent variable and the data obtained by Vicon as the dependent variable. Only statistically significant variables in regression analysis (*p* values < 0.05) are shown in bold. Moreover, the variables exceed thresholds (R^2^ > 0.25, ICC > 0.5) are underlined.

## Data Availability

The data presented in this study are available on request from the corresponding author. The data are not publicly available due to ethical restrictions.

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
