# Peer review of "Reliability and Validity Examination of a New Gait Motion Analysis System"

_sensors, 2025, doi:10.3390/s25041076_

Round 1
Reviewer 1 Report
Comments and Suggestions for Authors
This paper presents work evaluating VisionPose, a 2D markerless motion analysis system which is an alternative to traditional gait analysis methods. The study compares VisionPose to the gold-standard Vicon 3D motion capture system, investigating its potential as a more accessible and cost-effective solution for clinical settings.
The paper demonstrates merit. Through statistical methods and detailed comparisons of gait parameters, it validates VisionPose's capabilities and acknowledges its limitations. The system shows promise in measuring time-distance parameters, though it encounters challenges with more complex movements like tandem walking and transverse plane motion tracking according to the authors.
The manuscript's strongest contributions are in several key areas:
- The timely investigation of accessible alternatives to expensive marker-based systems, supported by robust statistical methods including ICCs and Cronbach's alpha
- A thorough analysis of various walking conditions and gait parameters, providing comprehensive insight into the system's capabilities
- The potential for impact in clinical settings through VisionPose's portable and cost-effective design
However, the work would benefit from several improvements. The research objectives and hypotheses need clearer articulation in the introduction. The relatively small sample size of 23 healthy adults limits the findings' generalizability to clinical populations. Please comment on this and suggest what power might be needed to be more generalizable for future work. The authors cite previous studies saying ‘generally 20 is enough’ essentially but this is tough to believe since there must be some more context for these statements. The study acknowledges VisionPose's reduced reliability for certain movements, it should explore how these limitations affect practical application in the clinic. Please comment on this.
In the results, particularly regarding joint angle changes in Figures 8-11, please add additional context and explanation in the text. Could the authors also add information about distances between markers which might be an indicator of underlying data quality.
The authors should comment at least briefly on a more detailed roadmap for future investigations. Perhaps, plans for testing VisionPose with pathological populations and improving its algorithmic capabilities for challenging movements. The manuscript should maintain more consistent terminology throughout, particularly regarding validity and reliability measures.
With revisions to address clarity, practical implications, and future directions, this work could influence the adoption of markerless gait analysis technologies or perhaps VisionPose itself in clinical and/or research settings.
Can the authors add more information about version number of VisionPose and the AI algorithm behind it – this is important, and also was the Imasen electric industrial company the creator of VisionPose and also the funder? Please be very clear about any potential COI.
Author Response
I apologize for the delay in replying.
Reviewer 2 Report
Comments and Suggestions for Authors
While the study is promising, several key revisions are needed to enhance clarity and depth in various sections of the manuscript. Below are my suggestions for revision:
- The Introduction section could further highlight the innovative aspects of VisionPose compared to other systems, such as Kinect and Vicon. Specifically, aspects like cost, ease of use, and accessibility could be emphasized to strengthen the rationale for using VisionPose in clinical applications.
- The Introduction should also include a brief introduction to the composite camera setup. Since this comparison is later discussed in the Methods section, providing an early explanation will help readers understand the significance of using both monocular and composite camera setups.
- The Methods section does not address the impact of the gender imbalance in the sample (with a significantly higher number of male participants). It would be beneficial to discuss whether this could introduce bias into the results and how this was accounted for in the study design.
- A brief introduction to tandem walking (Lines 138-142) should be added in the Methods section. This will help readers unfamiliar with the test understand its relevance, particularly in rehabilitation research, and why it was chosen as one of the gait tasks for this study.
- The manuscript does not address whether there were any restrictions on rest periods for participants between trials. Given that repeated data collection could lead to fatigue, which may affect the results, it would be useful to clarify whether rest was allowed and if so, how it was managed.
- Similar to the previous point, the manuscript does not mention whether the sequence of tasks was randomized or fixed. This could influence the results, especially if there is a learning effect or fatigue. Please clarify whether the order was randomized or kept consistent.
- Figures 4-6 show participants wearing tight-fitting clothes and barefoot. Was this the standard attire for all trials, or did participants wear different outfits for different trials? Uniform clothing is important when evaluating the applicability of the system in real-world, unmarked-point recognition scenarios, as clothing could potentially affect markerless motion capture.
- In the Methods section, particularly in Figure 3 or the corresponding description, it would be helpful to include the camera installation angles for the composite cameras. This information is crucial for the reproducibility of the results by other research teams.
- For clarity, I recommend changing the abbreviations for gait tasks in Figures 8 to 11 to their full names. This will avoid any potential confusion and ensure that the task names are clearly understood. Some recently studies could be added, such as: ‘Foot Morphology and Running Gait Pattern between the Left and Right Limbs in Recreational Runners’, Physical Activity and Health, 7(1), p. 43–52. Available at: https://doi.org/10.5334/paah.226.
- The limitations of plane gait analysis are acknowledged in the Discussion. However, there is room to expand on potential ways to address these limitations. It would be helpful to include some suggestions for improving the system’s ability to analyze multiple planes of motion, such as integrating additional camera angles or exploring three-dimensional analysis.
Reviewer 3 Report
Comments and Suggestions for Authors
Matsuda et al. evaluate the potential of VisionPose as an alternative to traditional marker-based systems like Vicon. The authors focus on comparing VisionPose’s performance using monocular and composite camera setups against the gold-standard Vicon 3D motion capture system, specifically for analyzing hip and knee joint angles and spatiotemporal gait parameters under various walking conditions. Although the subject is interesting and can potentially contribute valuable findings to the field, some issues must be addressed to align with the publication standards of sensors journal.
1. Can the authors explain how VisionPose’s markerless nature specifically reduces patient burden compared to Vicon beyond cost and portability?
2. Could the study include more diverse gait conditions or participants with movement impairments to evaluate generalizability?
3. How does the study account for potential biases introduced by manual intervention in data collection or processing?
4. Is there potential for integrating VisionPose with real-time clinical feedback tools, and if so, how could this be highlighted in the discussion?
5. What were the technical challenges encountered during composite camera setup, and how were they mitigated?
6. Several figures, such as Figures 2, 3, and 7, appear to have low resolution. Authors should re-export figures from the original software in higher resolution, at least 300 DPI.
7. There are redundancies, especially in the Results and Discussion sections. For example, ICC values and Cronbach’s alpha coefficients are repeatedly discussed. Consolidate these findings into a single, well-organized summary.
8. Authors should Perform a final language check for fluency and correctness, focusing on simplifying complex sentences and avoiding unnecessary repetitions. Also, ensure consistent use of past tense when discussing methods and results
Comments on the Quality of English Language
Perform a final language check for fluency and correctness, focusing on simplifying complex sentences and avoiding unnecessary repetitions.
Round 2
Reviewer 2 Report
Comments and Suggestions for Authors
All my questions have been well addressed, I recommend to accept.
Reviewer 3 Report
Comments and Suggestions for Authors
None Further comments required.